# A Practical Protocol for the Experimental Design of Comparative Studies on Water Treatment

**Long Ho [1],\*[ID], Olivier Thas [2,3,4], Wout Van Echelpoel [1][ID] and Peter Goethals [1]**

[1] Department of Animal Sciences and Aquatic Ecology, Ghent University, Ghent 9000, Belgium; Wout.VanEchelpoel@UGent.be (W.V.E.); Peter.Goethals@UGent.be (P.G.)
[2] Department of Mathematical Modelling, Statistics and Bioinformatics, Ghent University, Ghent 9000, Belgium; Olivier.Thas@UGent.be
[3] National Institute for Applied Statistics Research Australia, University of Wollongong, Wollongong 2522, NSW, Australia
[4] Interuniversity Institute for Biostatistics and statistical Bioinformatics, Hasselt University, Hasselt 3500, Belgium
\* Correspondence: Long.tuanho@UGent.be; Tel.: +32-926-438-95

**Abstract:** The design and execution of effective and informative experiments in comparative studies on water treatment is challenging due to their complexity and multidisciplinarity. Often, environmental engineers and researchers carefully set up their experiments based on literature information, available equipment and time, analytical methods and experimental operations. However, because of time constraints but mainly missing insight, they overlook the value of preliminary experiments, as well as statistical and modeling techniques in experimental design. In this paper, the crucial roles of these overlooked techniques are highlighted in a practical protocol with a focus on comparative studies on water treatment optimization. By integrating a detailed experimental design, lab experiment execution, and advanced data analysis, more relevant conclusions and recommendations are likely to be delivered, hence, we can maximize the outputs of these precious and numerous experiments. The protocol underlines the crucial role of three key steps, including preliminary study, predictive modeling, and statistical analysis, which are strongly recommended to avoid suboptimal designs and even the failure of experiments, leading to wasted resources and disappointing results. The applicability and relevance of this protocol is demonstrated in a case study comparing the performance of conventional activated sludge and waste stabilization ponds in a shock load scenario. From that, it is advised that in the experimental design, the aim is to make best possible use of the statistical and modeling tools but not lose sight of a scientific understanding of the water treatment processes and practical feasibility.

**Keywords:** experimental design; comparative studies; wastewater treatment; preliminary studies; virtual experiments; power analysis

## 1. Introduction

Water treatment is a crucial technology in water reuse and management. In the last century, technologies for drinking water purification and wastewater treatment have evolved rapidly to deal with the significantly increasing amount of water discharge as a result of the industrial revolution and the growth of urbanization [1]. To increase the removal and energy efficiency, environmental engineers have paid a great deal of attention to optimizing technology via comparing or contrasting different treatments. In particular, water treatments can be improved by changing five elements: (1) treatment technologies, (2) system configurations, (3) experimental methodology, (4) system inputs, and (5)

operational conditions (Figure 1). The improvement of the treatments can then be demonstrated via three sets of indicators, which represent the balanced sustainability of water technologies, including environmental performance, economic performance, and societal sustainability [2].

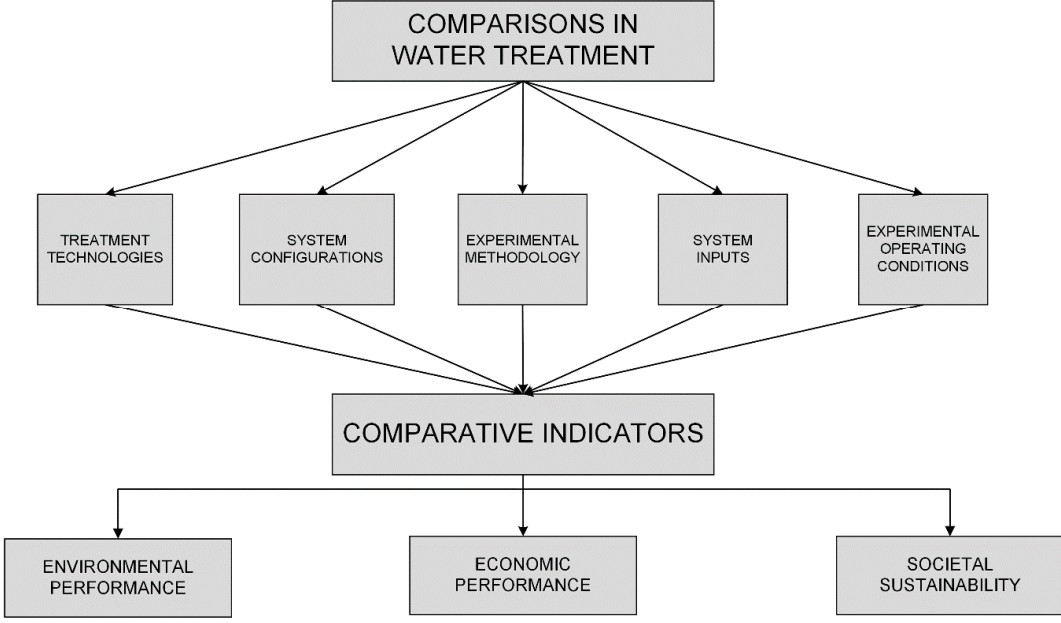

**Figure 1.** Overview of different types of comparative studies on water treatment and the use of comparative indicators.

Including multidisciplinary aspects, such as physics, chemistry, microbiology, and bioprocess engineering, studies on water treatment heavily depend on experimental observations [3]. In these experiments, three essential elements should be carefully considered: (1) analytical techniques, (2) experimental setup, and (3) statistical analysis. The first two elements have been well covered by many popular textbooks or guidelines, such as '*Standard Methods for the Examination of Water and Water*' of APHA [4], and '*Experimental Methods in Water Treatment*' of van Loosdrecht, Nielsen, Lopez-Vazquez and Brdjanovic [3]. On the other hand, to the authors' knowledge, there is no standardized procedure for statistical analysis in comparative studies on water treatment. Interestingly, in many other fields, such as ecology, evolution, biology, or behavioral science, researchers agree that experimental design and statistical analysis have an intimate link with each other. As such, it is crucial to carefully think about the exact formulation of the research questions, and explicitly translate them into statistical hypotheses before conducting experiments and collecting data [5].

This paper aims at providing a practical protocol for experimental design within the scope of comparative studies on water treatment. Throughout this protocol, we highlight key steps in the experimental design where the connection to statistics is most critical. Moreover, the role of modeling and simulation as the counterpart of real experiments in the virtual world is also emphasized in the protocol since these tools can be very effective but have rarely been used in experimental design. To keep the size of this paper manageable and avoid getting lost in the intricacies of statistics and modeling, we keep the protocol as simple as possible, hence, details on some basic principles of experimental design, such as blocking, randomization, replication, and factorial designs, which can be found in many textbooks, are not included. Finally, to ensure that experimenters find this protocol straightforward, easy to use, and highly applicable, its application is demonstrated via a case study on the performance comparison between conventional activated sludge (CAS) and waste stabilization pond (WSP) in a peak load scenario.

## 2. The Protocol

The main structure of the protocol is composed of four main stages with 10 modules and two feedback loops (Figure 2). Each of these stages will be discussed in the following sections, with specific attention to the section-specific modules.

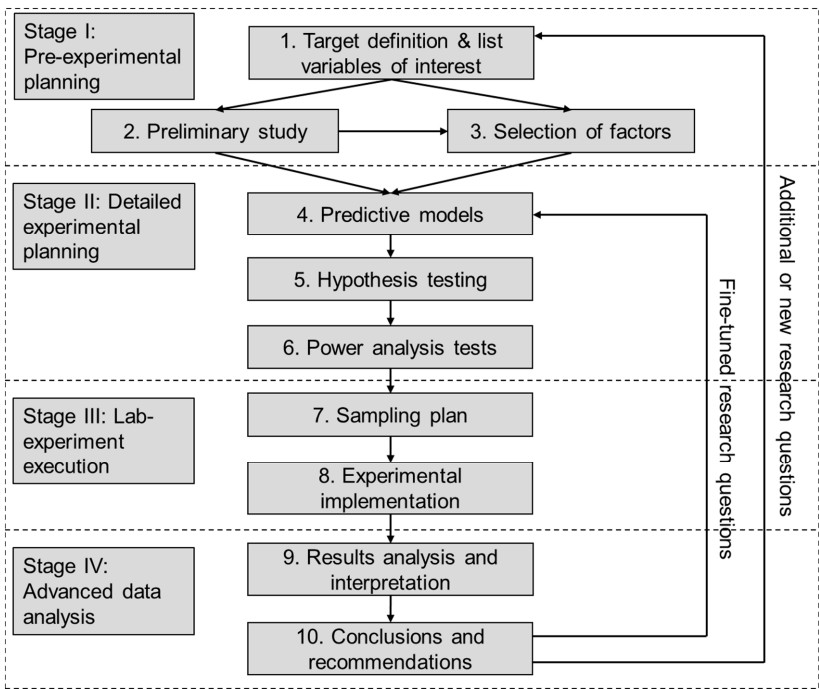

**Figure 2.** Main structure of the protocol. The protocol includes two feedback loops that allow for new research objectives or research questions, leading to further experiments.

### 2.1. Stage I: Pre-Experimental Planning

#### 2.1.1. Target Definition and a List of Variables of Interest

At the beginning, not only a clear and concise statement of the research problem but also a precise formulation of the research objectives has to be elaborated. Experimenters should think of themselves as managers who need to be clear about research goals in order to develop their respective action plans within the experiments. To do so, Doran [6] suggested that specific tasks should follow the S.M.A.R.T. approach: Specific, Measurable, Assignable, Realistic, and Time-related. As shown in Figure 1, types of comparisons and indicators can be addressed, whereas the list of response variables of the experiments can be drawn from the comparative indicators. The most common variables of interest in the field of water technology are treatment removal efficiencies with indicators of organic matters, nutrients, heavy metals, pathogens, or emerging contaminants. The number of response variables is decided based on both research objectives and resource availability, including money, time, and labor, which are required or already available for collecting and analyzing samples.

#### 2.1.2. Preliminary Study

Due to financial and time pressures on research, a preliminary study or pilot experiment is often ignored despite its important role in experimental design. First, it allows experimenters to get acquainted to treatment systems and analytical techniques. Thanks to the prior runs, the systems can operate more stably and accurately in the real experimental stage, especially when the experiments involve complex analytical techniques, advanced operational equipment, or bacterial incubation. Secondly, the information from preliminary studies can be used for making a sensible forecast of what can happen in full-scale experiments. This prediction is represented as a predictive model in step 4

(see further in 2.2.1). By reducing the chance of experimental failure, we can save more money and time. Furthermore, these pilot experiments also supply information about experimental error and the data variability necessary for sample size calculation in step 6 (see further in 2.2.3).

### 2.1.3. Selection of Design Factors

Design factors are parameters that can influence the performance of a treatment in general and response variables in particular. These parameters can be divided into five sub-categories, as shown in Figure 1, including technology types, configuration settings, experimental methods, system inputs, and operational conditions. By considering them and their interactions as potential design factors, factorial experiments can be created. Such design can serve several purposes. In a screening experiment, typically many factors are simultaneously evaluated, aiming at selecting the most important factors to be studied in detail. For screening, continuous parameters are discretized into a factor variable with two or three levels (e.g., "low" and "high," or "low," "median," and "high") and often only the linear effects are assessed. When the number of factors, $k$, is not too large, full factorial designs can be constructed, but note that the total number of experiments for a $2^k$ full factorial design grows exponentially with $k$ (e.g., with $k = 10$, $2^{10} = 1024$ or $3^{10} = 59,049$ experiments are required). Therefore, one may prefer to consider fractional factorial design. With fractional factorial designs, the total number of experiments can be reduced by a factor of 2, 4, or 8. However, this comes at a cost of not all effects being estimated, while estimable effects may show total confounding with other (interaction) effects. Care must be taken that the effects of the most interest are estimable and not confounded with other important effects. For each fractional factorial design, a list of estimable and confounded effects can be computed, which is referred to as the alias structure of the design. Textbooks, e.g., Quinn and Keough [7] and Montgomery [8], and software are available for assistance with setting up appropriate fractional factorial design. There have been special designs proposed to reduce the number of experiments, while still allowing the estimation of typical important effects (e.g., main, or linear, and first-order interaction effects), e.g., Plackett-Burman and Box-Behnken designs. Once the most important factors have been selected with screening designs, these factors can be studied with more precision in a follow-up experiment. The same designs can be used, but with a smaller number of factors, perhaps a full factorial design can be considered or a fractional factorial design with a larger fraction. Hence, more effects (e.g., interaction/nonlinear) can be estimated. Whenever possible, replications of experiments are advocated to further increase the precision or power of the statistical analysis.

In addition, designs can treat some factors as blocking factors, such as a season in which a series of experiments is performed or an operator executes experiments. The designs can be extended to block designs. Such designs are often categorized into complete and incomplete designs, or into balanced and unbalanced designs. A distinction can be made between easy-to-vary and hard-to-vary factors, e.g., a system configuration is hard to change, but experimental operating conditions are easier to vary. Split-plot designs take this distinction into account to create a feasible setup. Several extensions to even more complicated settings exist, e.g., split-split-plot and strip-split-plot designs.

Despite the large number of designs that have been described in the literature, their application is often not possible because of practical limitations, e.g., certain combinations of factor levels may not be possible in reality. By simply excluding these experiments from the design, the properties can be seriously affected, resulting in the loss of estimability of important factor effects. It is therefore not recommended to simply remove experiments from a design, unless the consequences are well known and understood. Instead, one should rely on more flexible design-generating methodologies. Adopting the D-optimality as a design selection criterion may be helpful. Combining notions of estimability for ensuring that a minimum set of possibly important effect can be assessed from the data and D-optimality for ensuring sufficient precision/statistical power, complicated designs that satisfy practical limitations may be generated [9]. However, these designs are computation-intensive, and an experienced statistician should be consulted. Especially noteworthy is that the method for data

analysis should correctly account for the design. For example, the analysis of data that come from a block design should always include the block as a factor in the statistical model, whether this block effect is significant or not. Excluding the factor from the model will result in invalid inferences on the other factors.

### 2.2. Stage II: Detailed Experimental Planning

#### 2.2.1. Predictive Models

Since failure is usually not an option in water treatment experiments due to their substantial costs, valuable results should be guaranteed by different means [10]. One of them is the application of predictive models that can predict the possible outcomes of the real experiments, allowing the generation of adequate research hypotheses besides the improved control of the real experiments. These models can be in many forms, such as mental, verbal, empirical, and mechanistic models. The last type is the most powerful and has extensive applications in the field of biological water treatment field. Based on the degree of conceptualizing and measuring biochemical and physical mechanisms, real-world experiments can be simulated via these models, seen as virtual experiments. To optimize and validate virtual experiments, data from preliminary studies can be a reliable source for model calibration and validation. Moreover, a good selection of design factors and their ranges bring the virtual-world models closer to real-world systems [11]. However, despite the substantial usefulness of these virtual experiments, they are substantially costly in terms of computation and time. In fact, similar to the real-world experiments, intensive quality assurance (QA) guidelines for virtual simulation models, including many time-consuming steps for model optimization and evaluation, i.e., sensitivity and identifiability analysis, model calibration, and uncertainty analysis [12]. From this perspective, empirical models can be an attractive option thanks to their simplicity with substantially lower requirement for available data and modest calculations for quickly obtaining reliable results at local scales [13].

#### 2.2.2. Hypothesis Testing

Statistical hypothesis testing is used to formally assess the effects of the changes in treatments on the response variable. It is a formalism of induction (from specific to general), which is based on the falsification theory of Karl Popper: a hypothesis can never be proven by finding confirmative evidence in sample data, but when evidence against a hypothesis is found in sample data, the hypothesis can be formally disproven. Hence, scientists have to set up experiments that allow them to find evidence against a well-stated hypothesis [14]. As such, statistical hypothesis testing can be seen as a computable implementation of Popper's paradigm that is accepted by the scientific community at large. In statistical hypothesis testing, the evidence in the data against a null hypothesis ($H_0$) is measured by a test statistic. Upon using probability theory, a null hypothesis is formally rejected when the test statistic exceeds a threshold, thereby controlling the probability of falsely rejecting the null hypothesis at a small fixed probability (the significance level, $\alpha$). The correct use of the probability theory, however, often requires assumptions on the distribution of the response variable. In fact, environmental engineers have frequently failed to check these assumptions. For example, a critical assumption is the independence of collected samples, which is frequently violated due to the dependency of collected data on time and space [15]. Particularly, longitudinal and/or spatial measurements taken close together in time and/or space show larger correlation than observations taken further apart. This phenomenon is known as autocorrelation. To correct the impact of spatiotemporal autocorrelations, mixed-effects modeling is highly advised as a valid statistical inference procedure as mixed-effects methods can incorporate the dependency through the introduction of random effects [16]. Other statistical approaches are also recommended, including autocovariate regression, spatial eigenvector mapping, autoregressive models, and generalized estimating equations [17].

### 2.2.3. Power Analysis Tests

A common misunderstanding in experimental design is that the sample size of an experiment should be as large as possible in order to obtain reliable results. However, from a certain sample size onwards, the marginal gain of further increasing the sample size becomes very small, hence performing too many measurements is a wasteful expense. On the other hand, an under-sized study can also be a waste of resources, because the underlying probability theory proves that there is little chance that sufficient evidence in the sample data will be found to reject the null hypothesis. Therefore, the determination of adequate sample size via power analysis is needed before running experiments [10]. The power of a statistical test is defined as the probability of correctly rejecting the null hypothesis ($H_0$). More specifically, in the alternative hypothesis ($H_a$), this probability is equal to $1 - \beta$ where $\beta$ as type II error rate is the chance of falsely retaining $H_0$ (Figure 3a). The value of power depends on three factors: (1) the significance level ($\alpha$); (2) the sample size and the variance of the experimental observations; (3) the true effect size [18]. Firstly, as the $\alpha$ and $\beta$ values are inversely related, increasing $\alpha$ decreases $\beta$ hence increases power (Figure 3b). The balance between $\alpha$ and $\beta$ is dependent on the research objective but $\alpha$ is normally set up smaller than $\beta$ because the consequences of false positive inference are considered more serious than those of false negative inference [19]. Figure 3c shows the relationship between sample size and the power using sample distributions under $H_0$ and $H_a$. When sample size increases, the variance of the sample mean is decreased, leading to the reduction of the overlap between the two distributions and, eventually, increasing the power. Lastly, as the third controlling factor, the true effect size is the difference between the two means in a two-sample *t*-test. When increasing the true effect size, the overlap between two distributions is decreased, resulting in a higher power as shown in Figure 3d. Based on these interactions, the required sample size for an experiment can be calculated. The specification of the variance is more difficult because no data are yet available in the design stage of the study. This is another reason for setting up a preliminary small scale experiment to obtain a more reliable estimate of the variance.

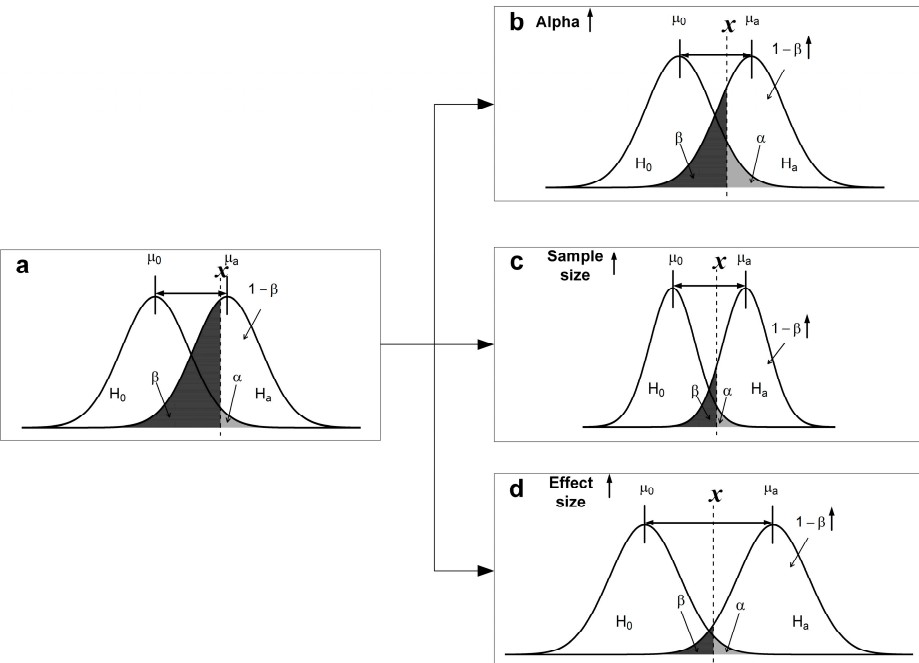

**Figure 3.** The effect of significant level ($\alpha$), sample size, and true effect size on the statistical power. Statistical power for one sample test for the mean of a normal distribution with know variance (**a**). The statistical power increases in case: (**b**) $\alpha$ increases or $\beta$ decreases; (**c**) sample size increases; (**d**) effect size increases [19].

*2.3. Stage III: Lab Experiment Execution*

2.3.1. Sampling Plan

If previous steps regarding pre-experiments and planning actions are properly implemented, generating an effective sampling plan is relatively straightforward. As a result of the power analysis, the sufficient number of samples can be calculated. The optimal sampling time and locations, together with required sampling frequency, can be derived from the experience from the preliminary study and the outputs of the predictive model [20]. The required time for each analytical test or experimental run can also be estimated from the experiences of the prior runs, which allows real experiments to be more controllable. As such, the experimenters can save time and money and avoid meaningless results.

2.3.2. Experimental Implementation

To ensure the experiment is efficiently executed according to the plan, it is important to monitor and control the running system and analytical tests. Experimenters must minimize the effect of noise factors or malfunctions to avoid experimental errors and sampling inaccuracies, leading to the reduction of outcome validity. As such, experimenters need to recognize the need for QA and commit their time and resources to develop and implement a comprehensive plan. Especially in case of water-treatment experiments, it is a challenge to develop a standardized method for experimental works that can be validated in different laboratories due to their undefined and interdisciplinary characteristics [3]. Therefore, quality control (QC) activities must be conducted in daily tasks of an experiment, i.e., collecting samples, performing analyses, checking running systems, processing and reporting experimental results. More importantly, experimenters must define a document-control procedure as a mean for data defensibility for each treatment and each analytical test [4]. The information from a logbook can be used as a reliable reference to identify and correctly interpret unpredicted results afterwards.

*2.4. Stage IV: Advanced Data Analysis*

2.4.1. Results Analysis and Interpretation

After data collection, instead of conventional visually comparing the values of the results, results can be analyzed by applying statistical data analysis, by which subjective conclusions can be avoided [8]. However, note that it is possible for an experiment to have a statistically significant difference but has little or no biological importance or practical value [21]. Indeed, rejecting a null hypothesis does not necessarily give a scientifically meaningful conclusion as in case of too small effect size, statistically significant difference does not mean practical significance. For this reason, a comprehensive interpretation should be a result of the combination between statistical analysis and scientific understanding about the treatments and processes.

For the presentation of results, graphical illustrations are often useful to attract audiences. Moreover, good graphics can bring many valuable information, such as content or structure of a dataset, the evaluation of statistical assumptions, and effective communication of results [22]. The results of statistical tests, such as regression equation or goodness-of-fit values, might be unnecessary to add to the figures to avoid complicated messages. Note that it is critical that the results should be presented as clearly as possible with their most important features, instead of submerging the audiences in an ocean of extraneous information [7].

2.4.2. Conclusions and Recommendations

After analyzing the results, environmental engineers must draw practical conclusions about treatment systems and recommend subsequent actions to decision makers. Note that by applying statistical analysis and modeling techniques, valid conclusions and recommendations are more likely to be withdrawn, hence, maximize the outcomes from the experiments and avoid generic conclusions.

Moreover, besides delivering scientific answers to the initial research questions, successful experiments can formulate new questions and hypotheses for future studies [10]. In fact, the experimentation can be considered as an iterative loop in which the feedback, including scientific understanding and statistical information, can be used to design more efficient subsequent virtual and real experiments.

*2.5. The Loops*

Throughout the protocol, it is crucial to keep in mind that experimentation is an iterative process for discovering the underlying mechanisms of the systems or processes. As such, sequential actions should be followed after the experiments, which are represented in the two feedback loops of the protocol. The first loop indicates the conventional progress of the experimental works in which the researchers tend to change the investigation area of design factors, add new or removal parameters, adjust system inputs, or change to different comparison approach by replacing with other response variable(s). This demonstrates the learning process in the field of the water treatment in which the optimal conditions or systems can be found in a specific situation after the comparative studies. From this perspective, it is recommended to not investigate a large portion of research resources on the first experiments or the beginning of a research. In fact, 25 percent of the total available budget is the maximum threshold for the first experiment, according to Montgomery [8]. Moreover, to prevent the failure or maximize the efficiency of the experiments beforehand, especially for the expensive and time consuming little-known systems and processes, mathematical models can deliver a valuable feedback for optimization. It is worth stressing that despite there always exists a trade-off in developing these preliminary predictive models as the more reliable information can be extracted from them the more resources and efforts have to be paid. These resources and efforts can originate from literature of previous similar studies or from the preliminary study in the stage of pre-experiment. As such, the measured data during the trial-run can be crucial in developing the data-driven models or in verifying the process-driven models. Depending on these purposes, the quantity and quality of data collection can be different [23]. The second loop represents this intimate connection between real experiments and virtual in silico experiments. This stage of evaluating the experimental data against the predicted model outputs can be described as system identification, which is a fundamental part of the process for developing new insights about the performance of systems or processes. This suggests an iterative procedure in which modeling plays a key role in addressing the knowledge deficiency, which is then handled with a new hypothesis-generating step and further experiments.

## 3. A Case Study: Performance Comparison of a Conventional Activated Sludge and a Waste Stabilization Pond in a Peak Load Scenario

The main objective of this case study is to illustrate the application of the aforementioned protocol as all steps of the protocol are clearly applied and described in each stage. From that, environmental engineers can find this protocol straightforward, easy to use, and highly applicable in water treatment studies.

*3.1. Pre-Experiments*

3.1.1. Step 1: Target Definition and a List of Variables of Interest

The most common application of water treatment technology, conventional activated sludge (CAS), is facing a criticism of having low cost-effectiveness and recovery potential, and high energy consumption [24]. On the contrary, waste stabilization ponds (WSPs) appear as a cheap but high effective treatment system [25]. The main objective of this study is to compare the environmental performance of these two wastewater treatment technologies. In particular, we wanted to evaluate and compare the removal efficiency and resilience capacity of the two systems in a shock load scenario with three phases: (1) initial phase; (2) high-strength-wastewater phase; (3) recovery phase, via two indicators, i.e., COD and TN. The overview of the experimental setup of two treatment systems is

illustrated in Figure A1 (Supplementary Material A) and its detailed description can be found in Ho et al. [26].

### 3.1.2. Step 2: Preliminary Study

Prior to the shock load scenario, a preliminary study was implemented within a period of about six months. During this start-up period, the CAS system and anaerobic ponds were inoculated with activated sludge from the Aquafin WWTP in Ossemeersen (Belgium) and two oxidation ponds were cultivated with an algal consortium. Other goals of this period were to: (1) get familiar with the system operation; (2) stabilize the two systems; (3) supply required data for calculating the removal rate of the preliminary models (see further in step 4); (4) obtain the essential information about the result variability for the power analysis tests in step 6.

### 3.1.3. Step 3: Selection of Design Factors

Based on the experience from running the preliminary study, controllable and uncontrollable factors were identified. In particular, air temperature was controlled at 21 ($\pm$2) °C and 16 hours of illumination per day-night cycle were automatically supplied via ordinary fluorescent lamps. In addition, pH (-), water temperature (°C), DO (mg $O_2 \times L^{-1}$), EC ($\mu$S $\times$ cm$^{-1}$), and sludge volume index (SVI) (mL $\times$ g$^{-1}$) were manually monitored on a daily basis. Some uncontrollable factors were identified, including the chemical contamination in the influent, the fluctuation of the peristaltic pump, and the variation of influent wastewater constituents. Although a great deal of attention was paid to limit their influence, these uncontrollable factors were the main reasons of the variations of the samples among the replicas.

### 3.2. Experimental Planning

### 3.2.1. Step 4: Preliminary Models

Two plug flow models were applied to predict the possible responses of the two systems to the shock load. In particular, the removal rates were calculated from the results of the start-up period, which were subsequently applied to estimate the pollutant levels in the effluent during the disturbance. These simulations were run for both COD and TN in advective-diffusive reactor compartment, whose hydraulic pattern is plug flow, in AQUASIM software [27]. These first-order models were chosen since they provide a good consensus between required resources and accuracy level in simulated results compared to basic empirical equations and complex mechanistic models [13]. In fact, the outputs of these models illustrated the duration of each phase and the simulated effluent concentrations of the two models as bell-shaped curves with three phases (Supplementary Material B).

### 3.2.2. Step 5: Hypothesis Testing

Four hypotheses were formed, based on the objectives of this study and the predictions of the first-order models (see Table 1). Since standard tests, including *t*-test or ANOVA F-test statistics, were not suitable because of the spatial autocorrelation between the samples collected within a reactor, likelihood ratio tests (LRTs) were applied in context of linear mixed-effects models (LMMs), in which the spatial autocorrelation was included as a random effect. On the other hand, two explanatory variables representing the fix effect were time and system, since these pollutant levels varied over time and their removal efficiencies were distinct from one system to another. After the model development, LRTs were employed to compare likelihood function values between these models and reference models, which were a linear regression with the same structure but without the random effect [28]. By comparing them, the necessity to include the random effect can be assessed. Especially noteworthy is that to compare the recoverability of each system in the last hypothesis, the random effect was employed to take into account the temporal autocorrelation of the samples collected within two phase,

i.e., before and after the peak. These statistical tests were carried out in R [29] using the nlme package with the lme function [30].

**Table 1.** Four null hypotheses with their relevant objectives.

| Null Hypotheses | Performance Comparison |
|---|---|
| $H_{0-1}$: The mean pollutant levels in the effluent of the two systems are equal during the beginning period. | Removal efficiency |
| $H_{0-2}$: The mean pollutant levels in the effluent of the two systems are equal during the shock load. | Resilience capacity |
| $H_{0-3}$: The mean pollutant levels in the effluent of the two systems are equal during the recovering phase. | Removal efficiency |
| $H_{0-4}$: The mean pollutant levels in the effluent are equal before and after the peak load. | Recoverability |

### 3.2.3. Step 6: Power Analysis Tests

We applied power analysis to determine the required sample size for each phase. To do so, we conducted a Monte Carlo simulation-based power analysis to find adequate sample size, which can result to a sufficient statistical power of 0.8. As such, different sample sizes of each phase were evaluated via simulations of different combinations, including 3–4 samples during the first phase, 4–8 samples during the peak load, and 4–5 samples during the recovery phase. Besides, the effect size representing the output variability was calculated, based on the results collected in the start-up period, ranging from 2 to 6 mg $\times$ L$^{-1}$ for both COD and TN. The $\alpha$ was set at 0.05. The simulation-based analysis was carried out in R [29]. The results of these simulations can be found in Supplementary Material C.

### 3.3. Experimental Conducting

#### 3.3.1. Step 7: Sampling Plan

Based on the results of power analysis tests, 16 samples were required to have the statistical power above 0.80, four samples during the first period, eight during the peak load, and four for the last phase. In case of the WSP system with an extensive HRT, five samples were needed during first extensive period, leading to three samples collected during the end phase. We scheduled the specific agenda for sample collection based on the experiences from the start-up run and the predictions of the first-order models.

#### 3.3.2. Step 8: Experimental Implementation

After the thorough preparation, the shock load scenario with three phases was conducted. During the first eight days, standard artificial wastewater was supplied to evaluate the removal efficiency. During five subsequent days of the high-strength-wastewater phase, the influent pollutant levels were increased by three times. The recovery of the systems was followed with the initial wastewater for the last 18 days. Note that QC activities were implemented intensively. Particularly, DO, pH, and EC probes were carefully calibrated and SVI was daily measured to assess the performance of the CAS system. A logbook reported all activities of each system was updated regularly.

### 3.4. Experimental Analysis

#### 3.4.1. Step 9: Statistical Analysis and Results Interpretation

The effluent concentrations of the systems during the peak scenario were demonstrated with the *p*-values of the statistical tests in Figure 4. In general, the two systems reacted in different patterns during the disturbance. For instance, during the high-strength-wastewater period, the removal efficiencies of AS for COD dropped 20%, while WSPs could still maintain low COD concentrations

in the effluent, leading to very low p-values (<0.001). The differences in the performance of two systems were more obvious in case of TN, with low *p*-values for each phase-based comparison between systems. Although the TN concentration of AS almost tripled during the peak, the systems were able to recover quickly within five days. On the other hand, WSP could keep their TN effluent concentrations low during the disturbance but were not able to recuperate to their initial values by the end of the experiment. This observation is supported by the significant difference obtained for TN concentrations in the initial and recovery phase within the WSP system ($p = 0.013$), while the difference within the AS system was not considered to be significant ($p = 0.089$).

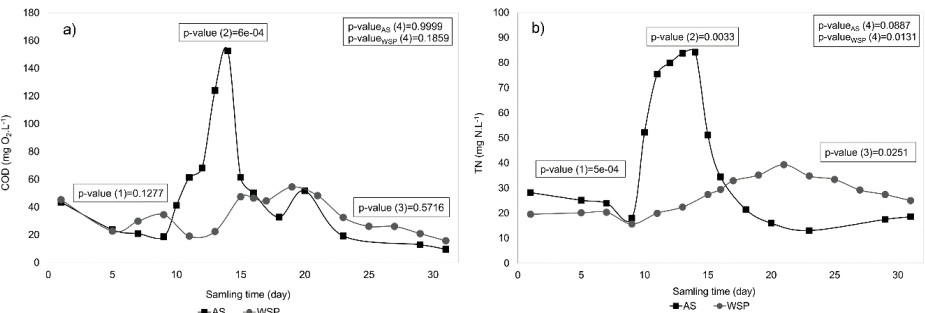

**Figure 4.** The effluent pollutant levels of the CAS and WSP with the *p*-value of the hypothetical tests during the peak scenario ((**a**): COD, (**b**): TN).

The predictions of the first-order models were compared with the experimental observations for model validation (Figure 5). Noticeably, in contrast to the relative accuracy of AS models, WSP models overestimated the effluent concentrations. The predicted effluent peaks started around 2.5 days later than the real peak, which can be a result of non-ideal hydraulic flow in WSPs (Figure 5C,D). Perhaps short-circuiting and dead zones in WSP systems could lessen the real HRT of the systems. Moreover, the removal rates of the plug-flow models were assumed to be constant but, in practice and our experiments, they were not [31]. For example, the increase of TN removal rate of WSPs during the peak period helped to maintain a low TN level in the effluent, causing the overestimation of the model predictions.

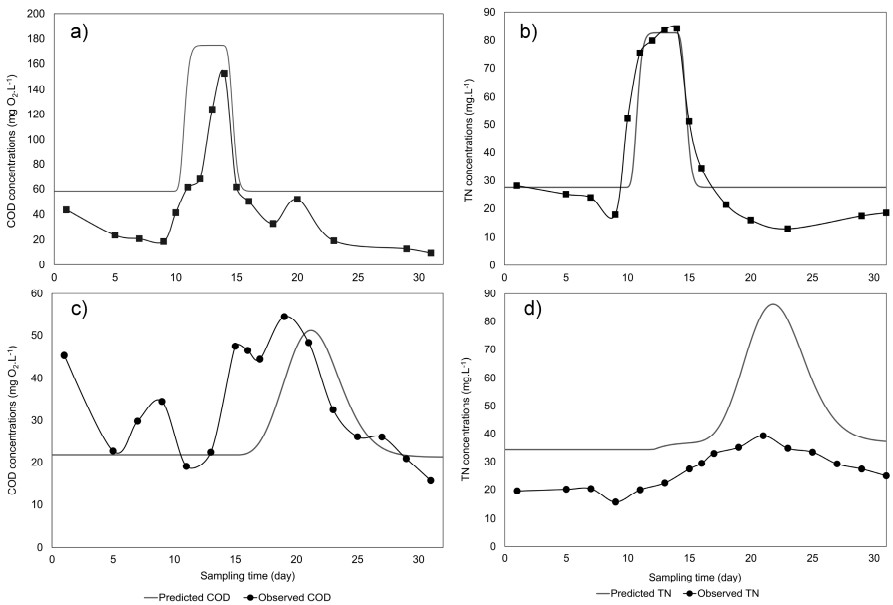

**Figure 5.** Predicted and observed effluent pollutant levels of conventional activated sludge ((**a**) and (**b**)) and waste stabilization pond ((**c**) and (**d**)) systems during the shock load scenario.

3.4.2. Step 10: Conclusions and Recommendations

From the results of the experiments, some practical conclusions and recommendations for the feedback loops can be drawn:

- Two systems appeared with relatively high capacity for removing organic matter (>90%), while higher nitrogen removal was obtained in WSP systems compared to AS;
- Regarding resilient capacity, the WSP systems proved their ability of replacing CAS in dealing with the shock load, especially regarding nitrogen removal;
- First-order kinetic models showed higher accuracy for CAS systems compared to WSP systems. A more sophisticated model is suggested for further studies, such as system optimization and performance analysis.
- To investigate the tolerance threshold for shock load of both systems, scenarios with higher strength of wastewater can be implemented in future experiments;
- To assess the sustainability of the two systems, other indicators regarding economic performance and societal sustainability need to be concerned in subsequent studies.

## 4. Discussion

The proposed protocol with its demonstration via a case study highlight the importance of incorporating powerful statistical and modeling techniques into experimental design thus valid and statistically sound results can be drawn. The crucial points are preliminary study, preliminary models, and power analysis for sample size determination. These steps are normally neglected in the experiments due to their requirement for time, money, and knowledge of statistics and modeling. However, as showed in the case study, their benefits strongly support the practicability and advisability of their implementation. For example, besides bringing many aforementioned advantages, the start-up period is especially necessary for biological experiments involving bacterial incubation. As the behavior of microorganisms depends on a number of internal mechanisms in response to changes in the environment [32], it normally takes quite a long time to be stable after being incubated in the new environment. A good example is a novel ammonium removal technology, anaerobic ammonium oxidation (anammox), which can require a start-up period of up to over a year, as reported in Van der Star et al. [33] and Nakajima et al. [34]. From this point of view, the necessity for prior runs in the experiments of physicochemical treatment processes can be less demanding because their removal reactions are much faster and controllable. However, given that useful information from the preliminary study can save a lot of frustration and disappointment due to failures of real experiments, the employment of a preliminary study is still highly recommended.

Another essential but often unnoticed tool for experimental design is a preliminary model, which performs in silico experiments to assess the behavior of the treatment systems. By applying mathematical algorithms to account for the experimental limitations, optimal conditions can be drawn, which is subsequently run in real systems hence time and money can be saved. To be valid in experimental design, it is important to note that preliminary models need to be optimized and evaluated first. Model optimization and evaluation are the link between virtual experiments and real experiments as they require real data to calibrate and validate the simulating models [35,36]. This data requirement is one of the major challenges of virtual experiments. Indeed, as water treatment systems are an interconnected web of many biochemical processes and reactions, their mechanistic models include not only multiple inputs but also many kinds of inputs, such as hydrodynamics, biochemistry, and microbiology. In other words, these complex models are nearly always overparameterized [37]. As such, empirical models can be a good option according to the parsimony principle of Spriet [38], in which the model should be less complicated than necessary for the description of the observed data. However, as a black-box approach, one should keep in mind that these data-based models only focus on system influent and effluent characteristics, resulting in poor predictive performance and significant underestimation of the uncertainty estimates hence are not globally predictive [39]. On the other hand,

mechanistic models can deliver high degree of understanding of biological and chemical processes happening in the treatment, hence they have a higher ability to apply them outside the boundary from which the model is developed [40]. Similar to the real-world experiments, intensive quality assurance (QA) guidelines for environmental modeling has been established to support the decision making process [41–43]. Particularly, uncertainty evaluation in integrated model is required following the precautionary principle in the field of water policy imposed by the EU Water Framework Directive [44]. Four main sources of uncertainty can be found in integrated environmental modeling, i.e., model input, model structure, model parameters, and model software [41,45,46]. These prior uncertainties can be quantified and propagated from model inputs to outputs via several methods, e.g., Gaussian error propagation, Monte Carlo simulation, and high-order Taylor expansion. A detailed description of these methods is beyond the scope of this paper, yet it is important to emphasize their variety and applications [47].

Experiments of water treatment are expensive, especially in studies on radiation, ultraviolet technology or molecular analysis [48]. Hence, it is important to ensure that the number of samples to be collected is sufficient but not wastefully excessive. Typically, experimenters in water treatment field defined the sample size via research experience and resource availability, without considering the power of the study. Consequently, the conclusions of many studies are more likely to be misleading and uninformative as a result of under- and overpowered studies [49,50]. To avoid that, textbooks [18,51] and specific software [52] have been useful tools for studies with simple statistical tests, such as comparing means via *t*-tests, ANOVA or tests related to linear regression models. However, since the experiments in water treatment studies are complex, essential assumptions, such as non-normally distributed or non-independently distributed data, are normally unsatisfied. Hence, numerical methods must be applied, as no simple formulas are available. In particular, as showed in the case study, Monte Carlo (MC) simulation procedures form a general and flexible framework that can be used for power and sample size calculations for complicated statistical methods [53]. However, despite these advantages, MC is rarely applied in the field of water treatment due to its complexity. To automate this process, a few packages have been developed in R [29], such as *pamm* [54], *clusterpower* [55], *longpower* [56], and *simr* [57].

This protocol underlines the multidisciplinary aspects of designing experiments, which involves preliminary laboratory work, modeling process, and power analysis. These tools not only maximize the amount of useful information for a given effort in the experiments, but also facilitate the understanding of treatment behavior and the interpretation of experimental results. Moreover, many multivariate statistic techniques of experimental design can be used for optimizing resources to reach certain goals of an experiment. For example, response surface methodology (RSM), a collection of mathematical and statistical techniques, is the most popular optimization method and is very useful for design and development of new systems or improvement of existing designs [58]. Based on the fit of experimental data and empirical models built via linear or square polynomial functions, the response of the system towards several design factors can be optimized [59]. As RSM is a sequential procedure, different stages of RSM application can be found. Starting at a remote point from the optimum on the response surface, least square method (LSM) in first-order model can be used to lead toward the vicinity of the optimum. Subsequently, a second-order model may be employed to detect the factor region in which optimum performance are reached [8]. Note that the second-order polynomial is not always well described by the non-symmetrical curvature response. For example, in case of the classic kinetic rectangular hyperbola of Monod equation, data logarithmic transformation is recommended before fitting to a second-order model [58]. Detailed demonstration of different second-order RSMs, i.e., Box-Behnken design, central composite design, Doehlert design, can be found in the book by Box and Draper [60]. Another technique with high practical merits is split plot designs. While factorial experiments tend to be large with more than two factors, split plot designs facilitate these multifactor factorial experiments by splitting them into two or more experimental units, i.e., whole plots and split plots [61]. Note that different from split plot designs dealing with crossed factorial factors, nested

designs focus on hierarchical structure of the experiments. As such, the main objective is to find the source of the variability of the response, which can be within blocks or between blocks. With the natural complexity of comparative studies on water treatment, the application of nested and split plot designs can be widespread. Having mentioned the statistical theory of experimental design in this protocol, it is important to bear in mind the purpose and use of these techniques as the use of too complex models and intricate statistics normally lead to a waste of time and effort.

## 5. Conclusions

A practical protocol particularly for experimental design of comparative studies on water treatment is presented with its demonstration via a case study. The protocol underlines the crucial but often neglected role of preliminary planning and statistical analysis on experimentation. As such, three key steps in the protocol, i.e., preliminary study, preliminary modeling, and power analysis, are strongly recommended to avoid the failure of the experiments leading to wasted resources and disappointing results. Despite being a time-consuming and labor-intensive process, the first two steps can provide very useful knowledge, experiences, and predictions on treatment performance thus, the experimenters can ensure the favorable outcomes of the experiments. Moreover, to avoid under- and overpowered studies causing misleading and uninformative conclusions, simulation-based power analysis is recommended for complex studies on water treatment. It should be emphasized that the pitfalls and assumptions of the statistical techniques must be considered before their application. Specifically, due to spatiotemporal autocorrelations of the data in the water treatment studies, their independence is violated, therefore invalidates the common tests, such as the *t*-test and ANOVA *F*-test. In this case, mixed-effects modeling is normally a good alternative as well as other statistical approaches, including autocovariate regression, spatial eigenvector mapping, autoregressive models, and generalized estimating equations. More importantly, the use of statistical tools, such as fractional factorial, response surface methodology, split plot, and nested designs, should be considered to produce the best possible experimental response. Lastly, environmental engineers should keep in mind that research experimentation are iterative processes in which sequential actions should be followed as feedback loops, allowing to generate new hypotheses and further experiments.

**Supplementary Materials:** The following are available online at http://www.mdpi.com/2073-4441/11/1/162/s1. Supplementary Material A: Schematic drawing of the two systems at lab scale. Supplementary Material B: Predicted effluent concentrations of activated sludge and waste stabilization pond systems during the peak scenario. Supplementary Material C: Results of power analysis tests.

**Author Contributions:** L.H. was involved in developing the protocol, experimental design and implementation, analyzing data, and writing the paper. O.T. participated in the experimental design and revising the paper. W.V.E. was involved in experimental design and implementation, and manuscript revision. P.G. was involved in experimental design and revising the paper.

**Funding:** This research was performed in the context of the VLIR Ecuador Biodiversity Network project. This project was funded by the Vlaamse Interuniversitaire Raad-Universitaire Ontwikkelingssamenwerking (VLIR-UOS), which supports partnerships between universities and university colleges in Flanders and the South.

**Acknowledgments:** We would like to thank Panayiotis Charalambous and Ana P-L Gordillo for their contributions during the experiments. We are grateful to WWTP Aquafin Ossemeersen for supplying the sludge needed in our experiments. Long Ho is supported by the special research fund (BOF) of Ghent University.

**Conflicts of Interest:** The authors declare no conflict of interest.

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
