# Peer review of "A Practical Protocol for the Experimental Design of Comparative Studies on Water Treatment"

_water, doi:10.3390/w11010162_

Round 1
Reviewer 1 Report
Although the main principles of preliminary stages, modelling and proper statistical analysis are well known to most researchers the paper makes a contribution by systematically presenting them in the form of a protocol.
I suggest considering instead of water treatment, wastewater treatment in the title (and other parts in the text) since both the theoretical and case study refers to wastewater treatment configurations
In lines 133-135 the same sentence appears twice. A minor spell check is required
Author Response
Cover letter
Manuscript ID: water-418539
A practical protocol for experimental design of comparative studies on water treatment by Long Ho, Olivier Thas, Wout Van Echelpoel, Peter Goethals.
Dear Reviewers,
We would like to thank you for relevant and constructive remarks. We have revised our manuscript accordingly. We acknowledge that these modifications definitely improve the quality of our manuscript. We hope that the changes and explanations are acceptable and satisfactory with the expectation of the editors and reviewers.
You can find in attachment the details of the modifications and explanations.
Thank you very much for revising our manuscript again!
Yours sincerely,
Long Ho
Department of Animal Sciences and Aquatic Ecology
Ghent University, Belgium

Reviewer 2 Report
This is an interesting paper, well suited for the Water readership. It basically reviews basic (more or less classic) statistical principles (both from the design and analysis points of view) and develops a simple protocol (formulated both in graphical and narrative senses) useful for practical work in water-related environmental engineering, discussing several practical and theoretical issues along the way. The ideas are well illustrated on a practical example carried from the design conception to the ultimate data analysis.
This paper should be published in Water.
The only thing that deserves additional work is the fact that despite the keywords “experimental design” appear in the paper name, there is absolutely no exposition to the statistical design theory in the text (apart from several instances of a reference to sources like Montgomery and few mentions of factorial design - without really explaining what it is).
This should be improved! The paper should contain at least a minimal discussion of the fact that there exists a huge statistical theory of experimental design which can optimize resources/effort for a given set of goals and that it is not only a dry theory but that it is also a very practical tool, indeed. In particular, simple designs like 2^n and 3^n factorials should be explained (mentioning also Box and Draper empirical optimization – response surface methodology) and then perhaps one or two examples of designs focused on efficiency for particular aspects with high practical merits - like fractional factorials or split plots.
Author Response

(The authors gave the same response as above.)
